# Quality of Information and Marketing of Rural Tourism Experience

**Marinês da Conceição Walkowski** [1] , **André Riani Costa Perinotto** [1,2,3,*] , **Vinicius Boneli Vieira** [2]
and **Anna Isabelle Gomes Pereira Santos** [3]

1    Tourism Department, Federal University of Paraná, Curitiba 80060-000, Brazil
2    Tourism Department, Parnaíba Delta Federal University, Parnaíba 64202-020, Brazil
3    Tourism Department, Ceará State University, Fortaleza 60714-903, Brazil
*    Correspondence: perinotto@ufpi.edu.br

**Abstract:** Investing in quality information contributes to the relationship between demand and suply. In this sense, this paper's objective is to analyze the quality of the information generated by the social media (Instagram) of the Acolhida na Colônia association. To identify the relevance of each attribute in the consumers' perception, categories and dimensions for quality information were analyzed based on the user's vision and semantic criteria. The main results revealed the difference in the quality of Instagram from the families who participated in the training. The quality of images and content of property's posts has also fed the Association's institutional Instagram. However, there is a need to expand the amount of information and constant updating.

**Keywords:** quality information; social media; Acolhida na Colônia

## 1. Introduction

A tourism destination is a physical space where visitors spend at least one night and includes tourist products such as support services and attractions. It has physical and administrative boundaries that define its management, image, and perceptions contributing to market competitiveness. Tourism destinations incorporate several stakeholders that generally include the local community and form a more extensive network of destinations [1].

The primary motivation for tourism in natural destinations is to break away from the routine and search for interaction with nature and residents. The landscape and existing relationships make this change possible and offer tourists an exchange of experiences.

The Brazilian countryside has changed, especially regarding relationships and forms of work. These changes enable farmers to increase their family income through new activities such as agrotourism, which aims to keep people in the countryside and improve their quality of life. In addition, the boundaries between rural and urban spaces have been reduced because the rural environment is adhering to activities previously common to urban spaces, seeking to streamline activities in the countryside [2–4].

In recent years, some tourism initiatives have taken place in both urban and rural spaces. They have helped to create employment and income for local communities, the need to value the local culture, diversify the offer of differentiated services, and from a history of resistance to mass tourism [5,6]. An example is the Acolhida na Colônia, an association formed by family farmers, part of the rural tourism segment, which encompasses the agrotourism activity in Brazil.

Investing in quality information aims to improve the commercial relationships between demand and supply in the municipality, based on the promotion, dissemination, and integration associated with the local tourist market. However, the SARS-CoV-2 coronavirus pandemic (COVID-19) demonstrated that the tourism industry is vulnerable, and this crisis has affected consumer behavior, requiring a new posture of destinations as products [7]. According to TRINET (TRINET is a list of international researchers in Tourism, who discuss

and debate current and emerging topics on tourism around the world. A closed group that uses emails for updates and discussions), an international discussion list comprised of researchers and tourism professionals, there are some trends and guidelines for the tourism industry, highlighting the marketing of destinations based on the following strategies: progressively expanding the market, targeting established markets first, and then expanding into new markets; focus on established markets (frequent customers); focus on the local industry of fairs and events; concentration, initially, on domestic and neighboring international markets; focus on social media—Instagram, Facebook, among others; and focus on business travel, being less discriminatory.

For this reason, analysis on social media (Instagram) was developed, as it is used as a relevant communication tool with the customer and has been a strategy to attract more visitors to tourism destinations in the countryside.

At the same time, analyzing the most indigenous tourism experiences developed in natural environments, as is the case of the Acolhida na Colônia, can highlight essential features, especially concerning its communication ability and quality information, aiming to attract more visitors. In this sense, the research objective was to analyze the quality of information generated by social media (Instagram) of the Acolhida na Colônia, based on the categories and dimensions proposed by Huang, Lee, and Wang [8].

In this sense, the purpose of this research was to analyze the quality information generated by the social media (Instagram) of the Acolhida na Colônia Agrotourism association based on the categories and dimensions proposed by Huang et al. [8].

## 2. Literature Review

### 2.1. Communication and Marketing in Tourism Destinations

The marketing in tourism destination should be used as a coordination mechanism and management that provides earnings for stakeholders and as a promotional tool [9,10].

Buhalis [11] states that the most successful tourism destinations will be those that take actions like establishing practical human resource training, cooperation with complementary and competing destinations in order to learn from them, and using innovation and marketing driven by research and technology to achieve four strategic objectives: (1) enhance long-term residents' prosperity; (2) delight visitors by maximizing their satisfaction; (3) maximize the profit of local companies and their multiplier effects; (4) optimize tourism impacts by ensuring a sustainable balance between economic benefits and socio-cultural and environmental costs [9]. In addition, destinations need to identify product attributes that will attract different segments of tourists and ensure that the promotional advertising represents a cohesive message [10,12] and an information linked to the emotion produced by the enjoyment of an experience [13].

For this reason, these same destinations do not offer a proper infrastructure to meet the tourist demand. This situation can be seen by the lack of signage, lack of information posts, inadequate access routes, inefficient use of landscape potential, and the lack of quality information in general, characterizing the lack of mentality regarding the tourism development [14].

It can be defined as a compilation of data numbered and ordered in a certain way and with a particular purpose [15].

The tourism activity uses the information to understand changes in customer behavior regarding their products and services, which is essential for communication in tourism marketing [16]. Advertising, promotion marketing, or even digital marketing actions have a limited reach of influence [17].

Technological innovations have influenced the habits and standards of the economy, politics, education, and especially communication. This is due to how information and new habits spread [18]. For example, technological advances and research in e-tourism [19] may contribute to the post-COVID-19 period, generating more information that will help change consumer behavior (risk perception, last-minute bookings, early bookings, and the

need for highly personalized trips) and likely trigger changes in the way to interact with people (from physical touch to voice or from input to automatic detection) [20].

*2.2. Quality Information*

The quality of information can be defined by the dimensions considered in the assessment and used to measure the quality. The dimensions are the properties of the qualities or the data characteristics [21,22].

The information availability and the environment conducive to business created on the internet allow travelers to know destinations, plan trips and itineraries, purchase tickets, choose and book hotels, purchase attractions, hire insurance, car rental, and others.

On the other hand, an organization's lack of quality information can have social and business impacts. Therefore, it must be analyzed, and efforts must be implemented for its solution [23].

Categories and dimensions for quality information were identified based on the user's view and semantic criteria according to Huang, Lee, and Wang [8] to identify the importance of each attribute in the consumers' perception.

Lee et al. [24] present the quality information by dividing it into four categories, intrinsic, contextual, representational, and accessibility quality. The categories have different characteristics which allow for evaluation. The intrinsic quality category refers to the quality information itself. The contextual category mentions that the information must consider the surrounding context. The representational and accessibility category deals with the system that stores and promotes access to information and how it is interpreted. Therefore, it must be easy to handle, consistent, accessible, and secure [24]. The information representation category concerns how it should be when available; that is, the information should be legible, easy to understand, and compatible. It can be said that the dimension that best represents this category is significance, as the level of understandability of the information is evaluated, and the accessibility category concerns the way of accessing information and is evaluated based on usability, security, and other dimensions. This last category evaluates the system where the information is stored and how it is stored and made available more than the information itself.

From the four categories mentioned above, Huang et al. [8] developed a list of fifteen dimensions for information analysis: (a) intrinsic quality—accuracy, objectivity, credibility, and reputation; (b) accessibility quality—access and security; (c) contextual quality—relevance, value-added, timeliness, completeness, and amount of data; (d) representational quality—interpretability, ease of use, concise representation, and consistent representation.

## 3. Research Methodology

*3.1. Experience Tourism Destinations—Acolhida na Colônia—SC*

The Acolhida na Colônia is an association formed by family farmers, part of the rural tourism segment, which encompasses the agrotourism activity. Agrotourism is an activity developed by organized family farmers willing to share their way of life, and cultural and natural heritage, maintaining their economic activities, and respecting the environment and local culture [25].

Through agrotourism, the tourism activity is carried out by family farmers active in agricultural and livestock activities and presupposes interaction and exchange of experiences with the visitor. However, it is noteworthy that in Brazil, the concept of agrotourism and rural tourism in family farming is often used interchangeably. The difference between them is that agrotourism presupposes farmers' organization (associative/group work, configuring itself as a community-based tourism initiative—CBT) for the activity development [25].

The Acolhida na Colônia Agrotourism association was founded in 1999 in Encostas da Serra Geral, in Santa Catarina, Brazil, based on the French Association Accueil Paysan, to provide quality of life and alternative income for family farmers in that location. Accueil Paysan is currently a worldwide network in 34 countries, intending to maintain family farmers [25].

Five years after its foundation, the group of farmers in the Encostas da Serra Geral began to stand out in the agrotourism activity, drawing the attention of the Santa Catarina State Tourism Department, which encouraged its expansion to other regions. As a result, Acolhida na Colônia raised funds from the Ministry of Tourism (MTur) and the Ministry of Agrarian Development (MDA) to make this expansion feasible. This expansion effort resulted in the consolidation of four more regional Acolhida na Colônia associations: Agrotourism Association Rota das Nascentes (2007), Vale dos Imigrantes (2007), Vale das Tradições (2007), and Serra Catarinense (2009).

Along the path of the Acolhida na Colônia, the institution received several awards (especially the Millennium Development Goals award) in recognition of its work, which increased its visibility and credibility, generating interest from other states, regions, and cities.

To understand how this research was done, it is essential to understand that the result of this second stage of expansion that consolidated two formal groups in two new States, Rio de Janeiro (a group of 10 families in the city of Casimiro de Abreu) and São Paulo (with a group of six families in the Parelheiros neighborhood). In Santa Catarina State, two new associations were consolidated, Encantos do Quiriri (2018), which covers the cities of Campo Alegre, Corupá, Rio Negrinho, and São Bento do Sul, in the northern Santa Catarina plateau. The Serra do Rio do Rastro association (2019) covers Lauro Miller's county in the southern region of Santa Catarina. In addition, the city of Alfredo Wagner, with ten families, joined the Serra Catarinense association in 2019. Six new counties are currently being implemented, Itapema, Camboriú, Florianópolis, Garopaba, Jaraguá do Sul, and Bom Retiro. All this was presented to contextualize and refer to tourism's theme, space, and activity in rural areas.

### 3.2. The Use of Social Media as a Marketing Strategy in the Study Site: Application and Experience

Part of the interest in this study came from one of the strategies adopted by the Acolhida na Colônia for advertising and marketing products and services. They held a training program about social media on 28–30 November 2019. During this period, nine properties of associated farmers from different regions were present. The training was given by the ATARé Communication and Education office and by two technicians from the Acolhida na Colônia association.

The training's objective was to provide the young farmers with some knowledge of basic photography and the use of Instagram since it is an essential communication tool with the customer. Social media make it very easy to establish a connection between the countryside and the city. Photography is an essential tool, and social media can be used to give these communities a voice.

The internet and social media have revolutionized all dimensions of social life: relationships, consumption, communication, and economy [19]. However, its use remains challenging for everyone, especially the tourism industry. The speed of technological change and participation in the global network force destination management and organizations to adopt and implement tools and techniques to improve the effectiveness and efficiency of destination marketing and to satisfy today's demanding tourist market [26].

The subjects covered in the training program were: introduction to photography, horizon, diagonal, borders, hands; centering, circle, triangle, photo selection, test post feed, caption, location, treatment; and metrics, interactivity, behavior tips, sequence of stories, among others.

The training was conducted using cell phone cameras, focusing on photographic composition and not just photographic technique.

Photography and Instagram have become relevant advertising tools, reaching many worldwide users. Instagram is a social media that prioritize publishing images and videos viewed and shared by users. Photography, in turn, is a visual code used to document, inform, and communicate, capable of motivating and influencing the tourist's decision-making process when choosing a destination [27].

The ATARé Communication and Education for Sustainability office developed the training with its methodology called Knowing to Love. This methodology has already been applied in Organic Oranges Cooperatives in the Rio Grande do Sul, Coffee Cooperatives in Minas Gerais, in the Family Farming Cooperative in Sergipe, and with groups of women farmers in Bahia. However, for each training, the methodology is adapted according to the target audience.

The Knowing for Love method mixes theory and practice, and the means used to develop the workshops are internet access, data shows, and cell phones. Theoretical contents are put into practice through practical exercises, and the results are analyzed in groups. The simple language and content are made from images, allowing people with low levels of education to access the information.

In addition, the topic related to digital marketing was also addressed through the relationship between the Acolhida na Colônia and the use of Instagram (number of posts, the purpose of stories and posts, to show the activities on the property, among others). Furthermore, they also addressed the benefits of digital marketing for the families, such as the visibility and reach of social media, and content related to the day-to-day in the countryside, e.g., cooking recipes, landscape, animals, family portrait, production, hosting infrastructure, and others. Finally, despite being little discussed, another relevant topic was storytelling by recording the property's daily life, understood as narrative.

Storytelling consists of reorganizing facts in order to make them more meaningful [28]. Thus, storytelling in tourism becomes a strategy to offer an outstanding tourism experience, as history transforms what would be an indifferent and unimportant space into an attractive tourism destination [29].

Instagram was the main app and social media presented to the families during the training were Instagram, given the migration of Facebook users to this media. The use of Instagram is also justified because it operates with images and less text, attracting more and more people.

The training workshop was aimed at farmers from the Acolhida na Colônia Agrotourism Association, Agreco (Agroecological Association of Encostas da Serra), and Cresol Encostas da Serra. Each agency mobilized their participants through targeted invitations.

The main results of the workshop demonstrated improvement in the quality of Instagram posts from participating families. The quality of images and content of the posts also contributed to increasing the information in the Acolhida na Colônia institutional profile on Instagram, such as the significant increase in followers. The ATARé monitors the content of the institutional profile.

Acolhida na Colônia also uses Instagram as a tool to advertise other initiatives, such as the delivery of the basket in Florianópolis city (an action to sell organic products called Da Horta à Mesa). The contents disclosed are related to agroecological production, hygiene, and protection measures against COVID-19 and favor the consumers' loyalty. The Association's interest in photographic images and the routine of posting on social media recent. However, it is recognized as an essential communication channel with tourists.

It should be noted that information is one of the primary economic resources for defining products and services to be offered. The quality of information can be defined by the cognitive authority quality factor for the user, based on what is perceived (credibility, influence, relevance, validity, reliability, and perceived value) and technical friendliness factors with the user, based on what is offered (form, accessibility, completeness, selectivity, time, flexibility, among others) [30,31].

Among the main difficulties of the workshop was the lack of suitable equipment (cell phone) for documentation was one of them, compromising the quality of the images. Furthermore, the reach of workshop participants was also limited (only 9 Acolhida's members participated), and it needs to be replicated to the other members to contribute to the institutional profile, which currently has a low number of linked properties to Instagram.

### 3.3. Methodology Steps

In this paper, the following steps were adopted as methods, a bibliographic research on communication and marketing and quality of information, the application of a semi-structured questionnaire, the participant observation with the Acolhida na Colônia Agrotourism Association, and the analysis of quality information and communication on the Association's social media (Instagram) based on categories and dimensions proposed by Huang et al. [8], and an application of a Likert-type scale.

The questionnaire comprises 14 semi-structured questions applied directly with the subjects linked to the technical team and assistance of the Acolhida na Colônia; it was made available online and sent by email. The questionnaire is one of the most used methods [32]; it consists of a list of questions formulated by the researcher to be answered by the researched subject.

To an analysis of the quality information in social networks was used to complement and better understand the tourism communication and marketing of the Acolhida na Colônia association, based on categories and dimensions proposed proposed by Huang, Lee, and Wang [8]. This process consists of four categories and fifteen dimensions, described in Table 1 below.

**Table 1.** Categories and dimensions of quality information analysis.

| Categories | Dimensions |
|---|---|
| Intrinsic | Accuracy: data is correct and reliable<br>Objectivity: data is unbiased and impartial<br>Believability: data is regarded as true and credible<br>Reputation: data is highly regarded in terms of its source or content |
| Accessibility | Accessibility: data is available, or easily retrievable<br>Access security: data is restricted appropriately to maintain its security |
| Contextual | Relevancy: data is applicable and helpful for the task at hand<br>Value-Added: data is beneficial and provides advantages from its use<br>Timeliness: data is sufficiently up-to-date for the task at hand<br>Completeness: data is not missing and is of sufficient breadth and depth for the task at hand<br>Amount of data: volume of data is appropriate |
| Representational | Interpretability: data is in appropriate languages and the definitions are clear<br>Understandability: data is easily comprehended<br>Concise representation: data is compactly represented<br>Consistent representation: data is presented in the same format |

The analysis of the quality of the information was carried out by five subjects participating in the research–three technicians from Acolhida na Colônia who taught the training course in communication and marketing, and two specialists in the area of tourism, communication and marketing. For the selection of these subjects, the non-probabilistic "snowball" sampling technique was applied [33], where the individuals selected to be studied invite new participants from their network of friends and acquaintances.

The analysis on social media took place on the Association's Instagram (in the properties participating in the survey), as it was the media used as a reference in the training of farmers and because it is considered an essential communication tool with the consumer public.

Initially, the grades applied by each interviewer and for each property were collected, but only eight were assessed. One property could not be evaluated, as it did not maintain an active profile on Instagram after the workshop. The data were organized by category and dimension lines and by columns bringing the scale score of each property.

A Likert-type scale was used to analyze the four categories and fifteen dimensions. The Likert scale assigns each statement a numerical value ranging from 1 to 5, also applied online.

Each respondent's final score is attributed to the sum of points they obtain in the alternatives. The respondents are asked to inform the degree of agreement/disagreement [33].

The Likert-type scale was used due to the need to hierarchize the amount of information acquired by the quality information analysis. The Likert scale allows for the scaling and qualification of the information analyzed, based on an already recognized and tested methodology. It also allows comparing what each subject who participated in the research understands about the quality information and communication analyzed in the Instagram profiles of each property.

## 4. Discussion and Results

*Quality Information on the Acolhida na Colônia Association's Social Media*

The properties selected for the research were: Pousada Vitória, Sítio Colina, Sítio OX Hill, Sítio e Café Hortêncio, Pousada Doce Encanto, Pousada Chalé Assing, Pousada Encanto Verde, Pousada Ekosol, and Pousada Maanaim, according to Figure 1.

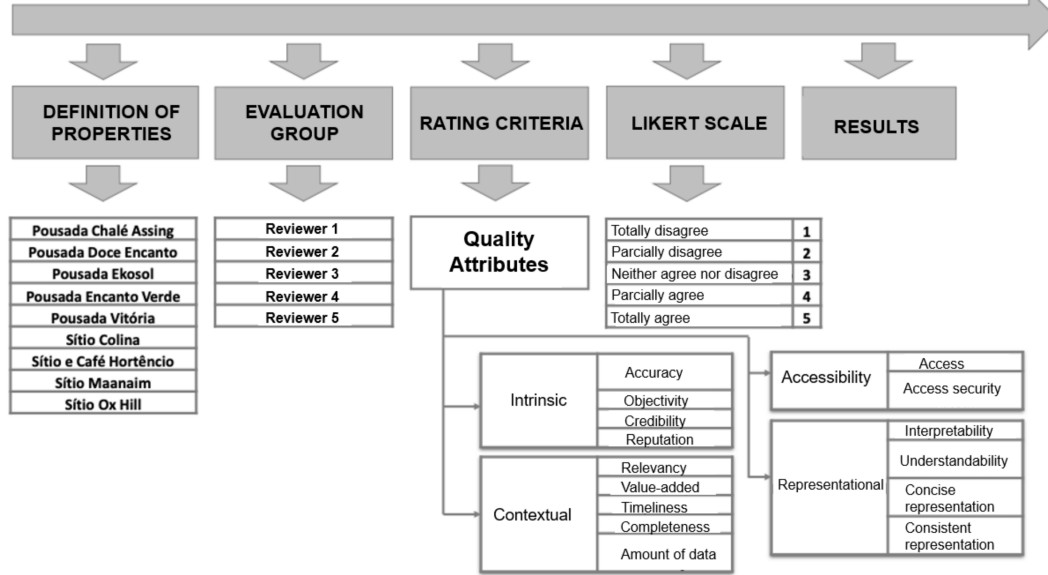

**Figure 1.** Process of analysis and evaluation of properties.

An analysis of the quality information in social media was used to complement and better comprehend the tourism communication and marketing of the Acolhida na Colônia association, based on categories and dimensions, proposed by Huang, Lee, and Wang [8]. This process consists of four categories and fifteen dimensions. A Likert-type scale was used for this analysis.

The quality information analysis on the social media of the Acolhida na Colônia association was made from 30 June to 5 July 2019. This analysis was carried out by five research subjects (three technicians from Acolhida na Colônia who taught the training course and two specialists in the field of tourism, communication and marketing).

The first data collection process was carried out from semi-structured interviews (available online and sent by email) that performed the social media training program.

Initially, the grades applied by each interviewer and for each property were collected, but only eight were assessed. One property could not be evaluated, as it did not maintain an active profile on Instagram after the workshop. The data were organized by category, dimension lines, and columns, bringing each property's scale score.

Of the 600 potential grades, 11 were not filled out and thus were not considered in the calculations, so there would be no modification in the averages. Next, an average of each property was assigned to each of the dimensions evaluated. Then, an average was assigned considering the sum of the eight properties (Table 2).

**Table 2.** Averages by categories and dimensions.

| Category | Dimensions | Pousada Chalé Assing | Pousada Doce Encanto | Pousada Ekosol | Pousada Encanto Verde | Pousada Vitória | Sítio Colina | Sítio e Café Hortêncio | Sítio OX Hill | Sítio Maanaim | Overall Average |
|---|---|---|---|---|---|---|---|---|---|---|---|
| Intrinsic | Accuracy | 5.0 | 4.8 | - | 5.0 | 5.0 | 4.4 | 4.8 | 4.0 | 4.8 | 4.73 |
| | Objectivity | 4.6 | 4.2 | - | 4.2 | 4.2 | 3.8 | 4.6 | 4.2 | 3.8 | 4.20 |
| | Believability | 4.8 | 5.0 | - | 5.0 | 5.0 | 5.0 | 5.0 | 4.4 | 4.8 | 4.88 |
| | Reputation | 5.0 | 4.4 | - | 4.6 | 4.4 | 3.8 | 4.6 | 4.2 | 4.2 | 4.40 |
| **Intrinsic Average** | | **4.85** | **4.60** | **-** | **4.70** | **4.65** | **4.25** | **4.75** | **4.20** | **4.40** | **4.55** |
| Accessibility | Accessibility | 4.0 | 3.6 | - | 3.8 | 4.6 | 4.2 | 4.8 | 3.6 | 3.2 | 3.98 |
| | Access security | 3.0 | 2.8 | - | 3.0 | 2.8 | 3.0 | 2.6 | 3.6 | 4.5 | 3.16 |
| **Accessibility Average** | | **3.50** | **3.18** | **-** | **3.40** | **3.70** | **3.60** | **3.70** | **3.60** | **3.85** | **3.57** |
| Contextual | Relevancy | 4.8 | 4.6 | - | 4.4 | 4.2 | 3.6 | 4.4 | 2.8 | 3.2 | 4.00 |
| | Value-Added | 4.2 | 4.8 | - | 4.4 | 4.6 | 4.0 | 5.0 | 3.2 | 4.0 | 4.28 |
| | Timeliness | 5.0 | 4.2 | - | 3.4 | 3.6 | 3.6 | 5.0 | 2.8 | 3.0 | 3.83 |
| | Completeness | 4.0 | 4.2 | - | 2.6 | 4.2 | 3.4 | 4.2 | 2.8 | 2.4 | 3.48 |
| | Amount of data | 4.2 | 3.0 | - | 2.2 | 3.4 | 2.2 | 3.6 | 1.8 | 1.8 | 2.78 |
| **Contextual Average** | | **4.44** | **4.16** | **-** | **3.40** | **4.00** | **3.36** | **4.44** | **2.68** | **2.88** | **3.67** |
| Representational | Interpretability | 3.8 | 4.3 | - | 4.0 | 4.6 | 3.8 | 4.5 | 2.3 | 3.5 | 3.83 |
| | Understandability | 5.0 | 4.6 | - | 4.6 | 4.6 | 4.0 | 4.4 | 3.0 | 3.6 | 4.23 |
| | Concise | 5.0 | 4.6 | - | 4.6 | 4.6 | 4.0 | 4.6 | 2.8 | 3.4 | 4.20 |
| | Consistent | 5.0 | 4.2 | - | 3.8 | 4.0 | 3.8 | 3.8 | 3.4 | 4.0 | 4.00 |
| **Representational Average** | | **4.70** | **4.41** | **-** | **4.25** | **4.45** | **3.89** | **4.33** | **2.86** | **3.63** | **4.06** |
| **Overall Average** | | 4.49 | 4.21 | - | 3.97 | 4.25 | 3.77 | 4.39 | 3.26 | 3.61 | |

The first analysis (Figure 2) obtained the final average of each of the properties of all parameters.

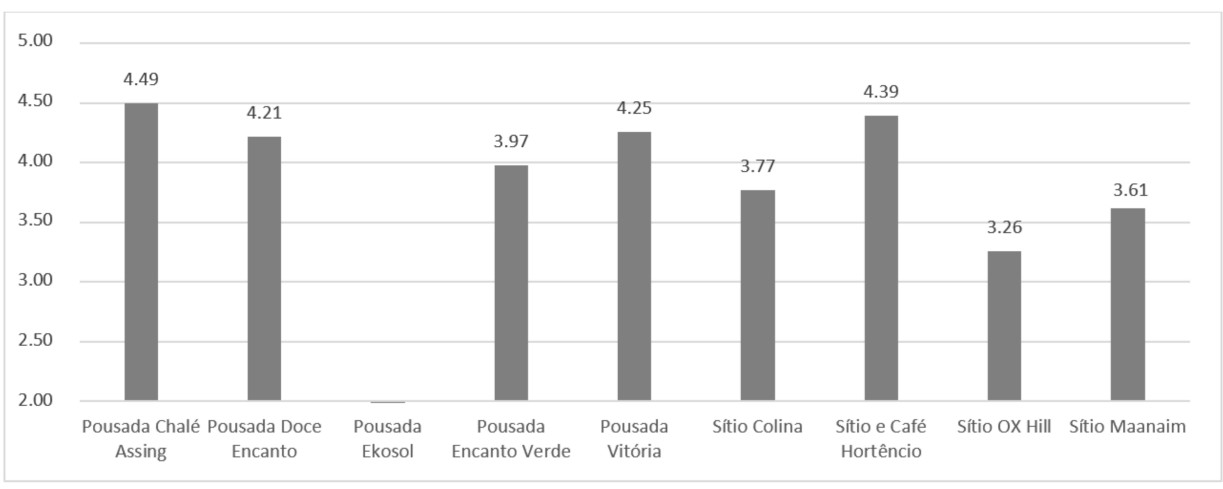

**Figure 2.** Final average of properties.

Observing the arrangement of properties in Figure 3 makes it possible to classify and define which ones need to improve information on their social media and which properties have a good quality of communication quality. A cutoff value (4) was used, where higher means are considered with quality and means below four needs improvement.

The choice of the cutoff score occurred due to the need for an agreement average (partial or total) so that the property evaluated is considered of good quality and means below this value (represented by "partially disagree", "total disagree", and "neither agree nor disagree") are considered properties that need to develop social media communication and marketing strategies.

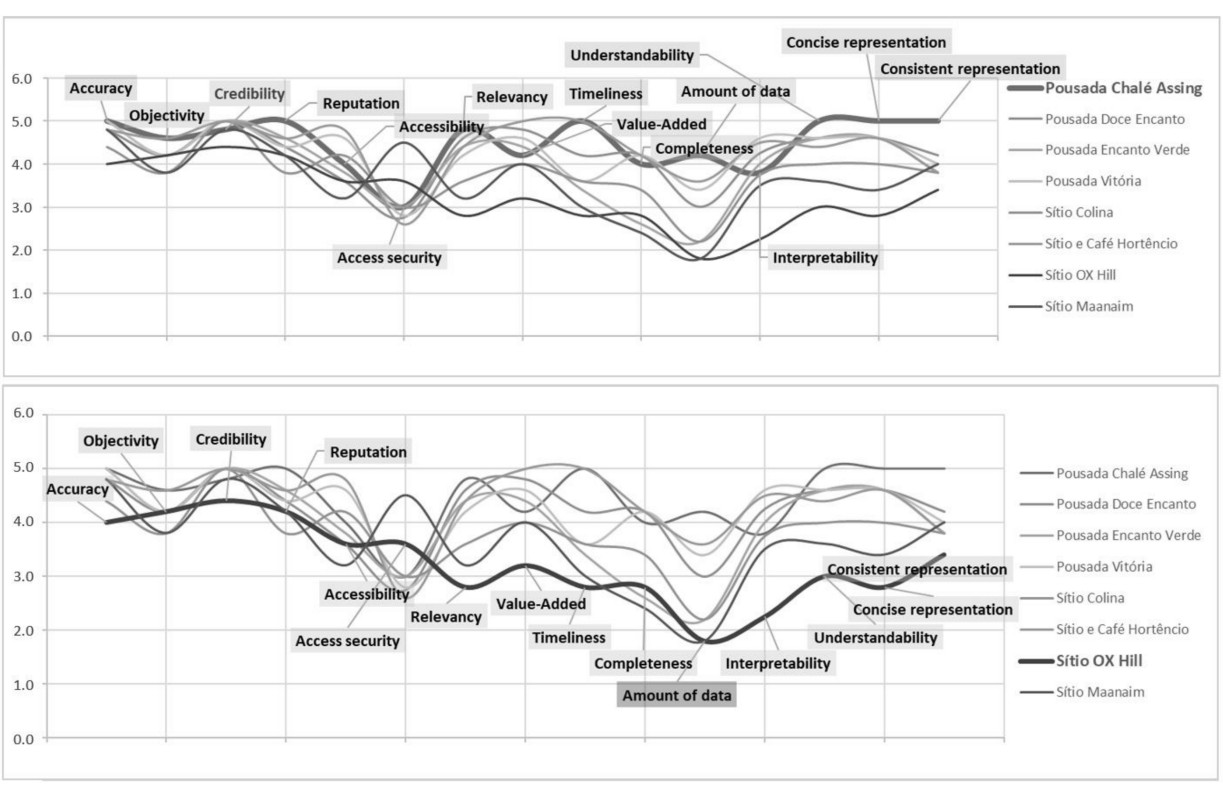

**Figure 3.** Representation by property.

After rating the properties, the analysis was deepened regarding the positive aspects and weaknesses (mean of the eight properties) to support new actions, such as training members to improve the marketing and communication of the properties continuously.

Then, the analysis shows the average of the four categories (Figure 4) to identify which stand out and need development.

The intrinsic and representational categories have the highest averages, between four and five, and the other categories (accessibility and contextual) have the lowest average. Thus, when analyzing the dimensions of each category, it is possible to observe the strengths and opportunities for improvement, especially in the accessibility and contextual dimensions.

The intrinsic dimension obtained the highest score (4.55), highlighting credibility with the highest rate (4.88), followed by accuracy (4.73). Therefore, the matters to be worked on are objectivity and reputation. Despite this, all dimensions varied between "partially agree" and "totally agree". The high scores for this category reveal a characteristic of rural establishments in Serra Catarinense, with a reliable source and content and accurate, correct, and reliable information. Credibility is, in fact, essential in tourist communication because tourists request information from relatives or friends as well as from people who have already been there before traveling [18,34]. Therefore, tourism destinations are strongly impacted, as credibility comes from the opinion of people who have experienced the same situations and difficulties, and the opinion of this social media user is exposed to the entire network, including their posted images [18]. Likewise, they found that these credible recommendations have more weight in decision-making than the information provided

by tourist prospectus, advertising in the mass media (press, radio, and television), tourist guides or articles, magazines, news, and reports for those who were exposed before the trip.

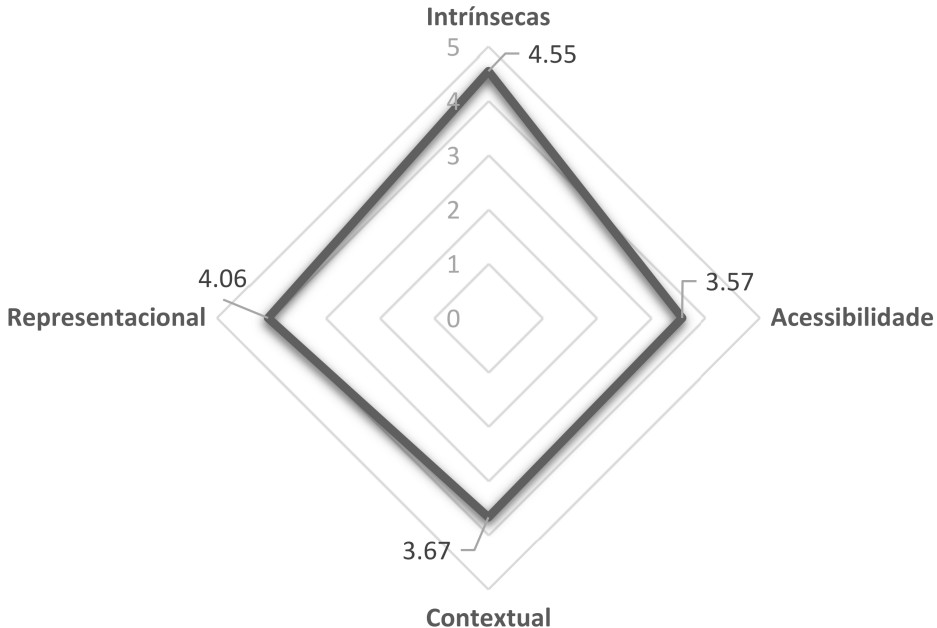

**Figure 4.** Average per category.

The second category analyzed is accessibility, which had the lowest average (3.57). This category needs to make more information available to the visitor with greater access control. The averages of the dimensions ranged between "disagree" and "partially agree".

The third category (contextual) had the most significant variation in the means. The dimensions of relevance and added value stand out, ranging between "partially agree" and "totally agree". Therefore, the information provided is valuable and beneficial, providing an advantage to the visitors. Timeliness (3.83) and completeness (3.48) had averages between "neither agree nor disagree" and "partially agree", demonstrating that keeping the information more up-to-date and not be lost is necessary. The lowest average belongs to the amount of data (2.78), between "partially disagree" and "neither agree nor disagree", demonstrating the need for a more significant amount of information available to visitors. This dimension shows the need for more significant concern in updating information, as according to Perinotto and Soares [18], tourism as an activity tends to be an intangible service, and the photographic image can be considered an attempt to make this service tangible. Tourism information sources are responsible for the destination presentation (or tourist service) and therefore can influence its attractiveness. According to [35], information is of fundamental relevance in tourism, as without it, the industry would not function. The tourist needs information to plan and decide before going on a trip. Following Alvares, Santos, and Perinotto [36], to reach the maturity of a tourism destination, it is necessary to permanently search for information that helps managers (public and private) to develop intelligent strategies that can meet the tourists' demand and, above all, surpass their expectations during the trip.

Finally, the representational category rated 4.06, but with lower variations in one of its dimensions. The interpretability dimension had the lowest average (3.83), revealing the need for greater attention to the language used. Despite this, the ease of understanding, highlighted in the concise and consistent representation demonstrates a good quality of information, so the problem is the form and not the content.

The characteristics of the tourism enterprises of the Acolhida na Colônia Association work as a parameter for data analysis; they are agriculture (main activity) and tourism (secondary activity) enterprises. The farmer's family often carries out advertising activities, with no structure or company managing the marketing actions. Therefore, some dimensions need improvements, such as insufficient information, temporality, accessibility, interpretability, completeness, and access security.

## 5. Conclusions

In this paper, qualitative and exploratory methods were adopted for the analysis. The research was developed in the following stages: bibliographic and secondary data on community-based tourism, communication, and marketing strategies, quality information, the questionnaire application, the participant observation with the Acolhida na Colônia Agrotourism Association, and the analysis of the quality information on the Association's social media (Instagram) based on categories and dimensions proposed by Huang, Lee, and Wang [8].

Tourism in rural spaces appears as a new job opportunity for these farmers through hospitality, food, or leisure services. Thus, their Instagram was analyzed, as it is used as an essential communication tool with the target audience and has been a strategy to attract more visitors to countryside destinations. Furthermore, the training methodology performed by the Acolhida na Colônia's team allowed to train the associates and impacted the quality information passed on to those seeking the properties' social media of properties participants of the project. Thus, the analysis methodology allows us to observe that the members were trained, impacting the quality information passed on to visitors.

Training with nine associated farmer properties from different regions was proposed. The topic related to the digital market was addressed through the relationship between Acolhida na Colônia and the use of Instagram (number of posts, the purpose of stories, feed posts, and the use of Instagram for marketing and not personal), benefits of digital marketing for families, such as the visibility and reach of social media, and content related to the routine in the countryside, cooking recipes, landscape, animals, family portraits, production, hosting infrastructure, among others.

Storytelling was recognized as a positive strategy in the dissemination of the daily activities of the properties through the farmers' narratives, but it is still little used by the families.

The results demonstrated a clear difference in the quality of Instagram between families who participated in the training and the ones who did not. The quality of the images and content of the properties' postings also contributes to strengthening the institutional profile of Acolhida na Colônia, with a significant increase in followers. Instagram has also been a tool for advertising initiatives like the basket delivery in Florianópolis, which reinforces the Acolhida na Colônia brand, with its agroecological production and short trade, and compliance with hygiene and protection protocols against COVID-19. The workshop assessment reinforced the relevance of the photographic image and the need for a routine of posts in the institutional profile to correspond with tourists.

Among the main difficulties was the lack of suitable equipment (cell phones) for documentation, compromising the quality of the images. Moreover, the workshop's scope was small (only nine members participated). Therefore, it needs to be replicated, as it makes all the difference, including the highlight given on the institutional Instagram (supported by a limited number of properties).

Regarding the analysis of the quality information on the Association's social media, the intrinsic and representational categories had the highest averages, between 4 and 5, and the other categories (accessibility and contextual) had the lowest average. Regarding the accessibility category, it is necessary to provide more information to the visitor and with greater access control.

The relevance and added value dimensions stand out in the contextual category, with averages between "partially agree" and "totally agree". Therefore, the information provided is valuable and beneficial, providing an advantage to the visitors. In the temporality and completeness dimensions, it was observed that it is necessary to keep the information more up-to-date to avoid being lost, as well as to invest in a more significant amount of information being made available to visitors.

Only the interpretability dimension had an average below four (4) in the representational category, demonstrating the need for more excellent care with the language used. Despite this, the ease of understanding, highlighted in the concise and consistent representation, demonstrates good quality information. So, the farmers should pay attention to the form of presenting the information.

Still, the photographs of landscapes, structures, and actions of the Acolhida na Colônia association boost tourism, inducing followers to visit the advertised destination. Besides, it makes people who access the rural enterprises' Instagram know and become interested in visiting these landscapes shown on social media. However, it is noteworthy that the farmer's own family often carries out the advertising activities, with no company managing the marketing actions. Therefore, some aspects need improvements, such as insufficient information, temporality, accessibility, interpretability completeness, and access security.

The potential of Instagram is also highlighted as a tool to indirectly advertise tourism destinations (especially in a location with such a unique rural methodology). Thus, the advisor team from Acolhida na Colônia, as well as those responsible for the city secretariats of tourism and agriculture, in possession of the data, should devise strategies to promote singular rural tourism and experience in the countryside. Furthermore, further research is needed on other companies in the tourism trade in order to infer how they treat their social media in the communication process with their current and potential customers.

This study demonstrates the importance of Acolhida na Colônia in generating employment and income for farming families. Training in technologies helps in the dissemination of its products through the quality of information, in the interactivity between the city and the countryside, and in the innovativeness of research.

It should be noted that no studies were identified that use the categories and dimensions proposed by Huang et al. in qualitative research, demonstrating potential for new categories and analyses.

Finally, information is one of the defining economic resources for products and services offered by the Acolhida na Colônia Association. The quality of this information can be supported by the credibility and reliability acquired over more than 20 years since the Association was established.

**Author Contributions:** Conceptualization, M.d.C.W., A.R.C.P. and V.B.V.; methodology, M.d.C.W., A.R.C.P. and V.B.V.; validation, M.d.C.W., A.R.C.P., V.B.V. and A.I.G.P.S.; formal analysis, M.d.C.W., A.R.C.P. and V.B.V.; investigation, M.d.C.W., A.R.C.P. and V.B.V.; resources, M.d.C.W.; data curation, M.d.C.W., A.R.C.P. and V.B.V.; writing—original draft preparation, M.d.C.W., A.R.C.P. and V.B.V.; writing—review and editing, M.d.C.W., A.R.C.P., V.B.V. and A.I.G.P.S.; visualization, M.d.C.W., A.R.C.P., V.B.V. and A.I.G.P.S.; supervision, A.R.C.P.; project administration, M.d.C.W. All authors have read and agreed to the published version of the manuscript.

**Funding:** This research received no external funding.

**Institutional Review Board Statement:** Not applicable.

**Informed Consent Statement:** Not applicable.

**Data Availability Statement:** No applicable.

**Conflicts of Interest:** The authors declare no conflict of interest.

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
