# Peer review of "Quality of Information and Marketing of Rural Tourism Experience"

_knowledge, doi:10.3390/knowledge2030025_

Round 1
Reviewer 1 Report
The paper set out to analyze the quality of information provided on the Instagram platform by members of the the Acolhida na Colônia Association. However, as currently written, the paper requires much guessing at what the authors are trying to communicate. This is in all likelihood due to poor translation as well as common mistakes made in English. Three examples to illustrate this comment:
1) l. 41-44: It is noteworthy that in recent years, some tourism initiatives have taken place in both spaces (urban and rural) and have helped to generate employment and income for local communities and the need to value local culture, diversify the offer of differentiated services, and from a history of resistance to mass tourism.
2) l. 89-90: The tourism activity is developing in an incipient way in several destinations and the lack of planning of the territory.
3) l. 113-114: early bookings desires in new contexts such as museums,
The literature review is neither well organized nor well summarized. For instance, the section on "Communication strategies and marketing of experience" does not explain what these strategies are nor how marketing of an experience differs from that of services. Instead, it addresses planning issues such as poor infrastructure. The economy, politics, etc. do not have "habits". The literature should also be searched for relevant articles published in the last two years.
Both the explanation as to the choice of the Acolhida na Colônia Association and the training that occurred should be moved up as they provide context and are not part of the methodology. Both need to be more explicit and succinct. There should also be a section on the Instagram platform, its use in Brazil and by businesses, especially those that are members of Accueil Paysan in Brazil and more specifically the Acolhida na Colônia association (is this different from the Acolhida na Colônia Agrotourism association?).
As far as the methodology is concerned, it is poorly explained. It would appear (based on Figure 1) that the subjects mentioned in the methodology section are actually the reviewers of the Evaluation Group, whereas there is no discussion of how the properties were chosen. How many are there in total? How many participated in the training? Were these the only ones considered? There seems to be a suggestion that Instagram accounts of properties that participated in the training as well as those that did not were analyzed, but the analysis and discussion does not seem to distinguish between them. There is also reference to participant observation but it is not clear what role this played in the discussion. Similarly, the discussion mentions a pretest but this is not addressed in the methodology.
Reviewer 2 Report
Dear Authors,
Clarity and precision are amongst the most important criteria of an article. Different notions were used for the same thing, such as: tourist destination (line 22, 75,..) or tourism destination (line 26,..) (make your choice), rural spaces, municipality, countryside, natural environments.
Lines 61-62 - behavior changes were mentioned. Which is the relationship between this concept and the subject of the article? It would be appropriate to explain the importance for introducing this concept in the article.
Only at line 72 the reader understands what is the association mentioned from the very beginning of the article. Previously, only fragmented information were presented (Abstract, line 64,..). In the line with the ambiguity problems, please explain what is Acolhida na Colônia: a municipality, a region, a rural space, an agrotourism association or else?
Lines 89-91. Please clarify the relevance of the planned development process and infrastructure of a destination for the subject of your article: Instagram's images and content. Literature review section could be more focused on communication issues and information quality, i.e. especially social media platforms in shaping a destination’s future development.
The Research Methodology section mainly describes the association, its members and their training program, elements which could be included in another section of the article.
The Research Methodology section is ambigous. Who are the respondents? What is the relevance of participants’ observation (line 179)? In conclusion, authors present one or several researches? Moreover, a qualitative research does not provide average values to be further analyzed.
The Conclusion section is incomplete. The main elements that were not discussed by authors were the relevance of their findings and comparisons of their results with other findings on the same subject.
Round 2
Reviewer 1 Report
This revised version shows a significant improvement over the first version submitted. Language is also much better, although there are still some problematic sentences like this incomplete one on p. 2, l. 89-90: “The tourism activity is developing in an incipient way in several destinations and the lack of planning of the territory [15,16].” or this very awkward sentence on p. 3, l. 118-120: “The quality information can be defined by the set of dimensions considered in the assessment and used to measure the quality, since, according to the literature, the dimensions are the properties of the qualities or the data characteristics [27,28].” Please also note that “data” is the plural of datum.
Although the revised heading under 2.1. Communication strategies and marketing of experience in tourist destinations is better, the section still devotes a paragraph to planning. While very important, it might deserve its own heading because it does not fit under the current one. The paper credits Smith and Hanover (2016) with experience marketing as a new marketing method but Pine and Gilmore discussed this already in 1998, and it was predicted by Toffler in his 1970 book Future Shock.
The Acolhida na Colônia and the training that was conducted are still not properly introduced. The sections 3.1 and 3.2 could be moved up to the beginning of the methodology section with a new 3.3 addressing methodological steps including data collection. This would provide the necessary context to understanding the process.
Reviewer 2 Report
Dear Authors,
I appreciate the present form of your paper.
But, I found inconsistencies between your Cover letter and the present paper.
In the Cover letter you mentioned the use of the concept "tourism destination" in the paper. Unfortunatelly, it is not the only one used. For exemple, at lines 22,75, 78,.. the term "tourist destination" was used.
Between lines 22-24, the two phrases contradict to each other. First, you mention that a tourist spends at least one night (which means that the tourist spends at least one night in an accommodation unit), and after that you mention that it is about one-day return trip – which means that a tourist does not spend the night at the destination. One-day return trip does not include accommodation. Please clarify these statements.
The phrase between lines 75-77 is difficult to understand, and in its present form it is also incorrect. The same is valid for the two phrases between lines 89-91, lines 124-126.
In the Cover letter you mentioned that at lines 720-726 two phrases were added. But the last line of your paper is 532.
